# Distributed Estimation, Information Loss and Exponential Families

**Qiang Liu**          **Alexander Ihler**
Department of Computer Science, University of California, Irvine
qliu1@uci.edu      ihler@ics.uci.edu

## Abstract

Distributed learning of probabilistic models from multiple data repositories with minimum communication is increasingly important. We study a simple communication-efficient learning framework that first calculates the local maximum likelihood estimates (MLE) based on the data subsets, and then combines the local MLEs to achieve the best possible approximation to the global MLE given the whole dataset. We study this framework's statistical properties, showing that the efficiency loss compared to the global setting relates to how much the underlying distribution families deviate from full exponential families, drawing connection to the theory of information loss by Fisher, Rao and Efron. We show that the "full-exponential-family-ness" represents the lower bound of the error rate of arbitrary combinations of local MLEs, and is achieved by a KL-divergence-based combination method but not by a more common linear combination method. We also study the empirical properties of both methods, showing that the KL method significantly outperforms linear combination in practical settings with issues such as model misspecification, non-convexity, and heterogeneous data partitions.

## 1  Introduction

Modern data-science applications increasingly require distributed learning algorithms to extract information from many data repositories stored at different locations with minimal interaction. Such distributed settings are created due to high communication costs (for example in sensor networks), or privacy and ownership issues (such as sensitive medical or financial data). Traditional algorithms often require access to the entire dataset simultaneously, and are not suitable for distributed settings.

We consider a straightforward two-step procedure for distributed learning that follows a "divide and conquer" strategy: (i) local learning, which involves learning probabilistic models based on the local data repositories separately, and (ii) model combination, where the local models are transmitted to a central node (the "fusion center"), and combined to form a global model that integrates the information in the local repositories. This framework only requires transmitting the local model parameters to the fusion center once, yielding significant advantages in terms of both communication and privacy constraints. However, the two-step procedure may not fully extract all the information in the data, and may be less (statistically) efficient than a corresponding centralized learning algorithm that operates globally on the whole dataset. This raises important challenges in understanding the fundamental statistical limits of the local learning framework, and proposing optimal combination methods to best approximate the global learning algorithm.

In this work, we study these problems in the setting of estimating generative model parameters from a distribution family via the maximum likelihood estimator (MLE). We show that the loss of statistical efficiency caused by using the local learning framework is related to how much the underlying distribution families deviate from full exponential families: local learning can be as efficient as (in fact exactly equivalent to) global learning on full exponential families, but is less efficient on non-exponential families, depending on how nearly "full exponential family" they are. The

"full-exponential-family-ness" is formally captured by the *statistical curvature* originally defined by Efron (1975), and is a measure of the minimum loss of Fisher information when summarizing the data using first order efficient estimators (e.g., Fisher, 1925, Rao, 1963). Specifically, we show that arbitrary combinations of the local MLEs on the local datasets can approximate the global MLE on the whole dataset at most up to an asymptotic error rate proportional to the square of the statistical curvature. In addition, a KL-divergence-based combination of the local MLEs achieves this minimum error rate in general, and exactly recovers the global MLE on full exponential families. In contrast, a more widely-used linear combination method does not achieve the optimal error rate, and makes mistakes even on full exponential families. We also study the two methods empirically, examining their robustness against practical issues such as model mis-specification, heterogeneous data partitions, and the existence of hidden variables (e.g., in the Gaussian mixture model). These issues often cause the likelihood to have multiple local optima, and can easily degrade the linear combination method. On the other hand, the KL method remains robust in these practical settings.

**Related Work.** Our work is related to Zhang et al. (2013a), which includes a theoretical analysis for linear combination. Merugu and Ghosh (2003, 2006) proposed the KL combination method in the setting of Gaussian mixtures, but without theoretical analysis. There are many recent theoretical works on distributed learning (e.g., Predd et al., 2007, Balcan et al., 2012, Zhang et al., 2013b, Shamir, 2013), but most focus on discrimination tasks like classification and regression. There are also many works on distributed clustering (e.g., Merugu and Ghosh, 2003, Forero et al., 2011, Liang et al., 2013) and distributed MCMC (e.g., Scott et al., 2013, Wang and Dunson, 2013, Neiswanger et al., 2013). An orthogonal setting of distributed learning is when the data is split across the variable dimensions, instead of the data instances; see e.g., Liu and Ihler (2012), Meng et al. (2013).

## 2  Problem Setting

Assume we have an i.i.d. sample $X = \{x^i : i = 1, \ldots, n\}$, partitioned into $d$ sub-samples $X^k = \{x^i : i \in \alpha_k\}$ that are stored in different locations, where $\cup_{k=1}^d \alpha_k = [n]$. For simplicity, we assume the data are equally partitioned, so that each group has $n/d$ instances; extensions to the more general case is straightforward. Assume $X$ is drawn i.i.d. from a distribution with an unknown density from a distribution family $\{p(x|\theta) : \theta \in \Theta\}$. Let $\theta^*$ be the true unknown parameter. We are interested in estimating $\theta^*$ via the maximum likelihood estimator (MLE) based on the whole sample,

$$\hat{\theta}^{\mathrm{mle}} = \arg\max_{\theta \in \Theta} \sum_{i \in [n]} \log p(x^i|\theta).$$

However, directly calculating the global MLE often requires distributed optimization algorithms (such as ADMM (Boyd et al., 2011)) that need iterative communication between the local repositories and the fusion center, which can significantly slow down the algorithm regardless of the amount of information communicated at each iteration. We instead approximate the global MLE by a two-stage procedure that calculates the local MLEs separately for each sub-sample, then sends the local MLEs to the fusion center and combines them. Specifically, the $k$-th sub-sample's local MLE is

$$\hat{\theta}^k = \arg\max_{\theta \in \Theta} \sum_{i \in \alpha^k} \log p(x^i|\theta),$$

and we want to construct a combination function $f(\hat{\theta}_1, \ldots, \hat{\theta}_d) \to \hat{\theta}^f$ to form the best approximation to the global MLE $\hat{\theta}^{\mathrm{mle}}$. Perhaps the most straightforward combination is the linear average,

$$\textit{Linear-Averaging}: \quad \hat{\theta}^{\mathrm{linear}} = \frac{1}{d}\sum_k \hat{\theta}^k.$$

However, this method is obviously limited to continuous and additive parameters; in the sequel, we illustrate it also tends to degenerate in the presence of practical issues such as non-convexity and non-i.i.d. data partitions. A better combination method is to average the *models* w.r.t. some distance metric, instead of the *parameters*. In particular, we consider a KL-divergence based averaging,

$$\textit{KL-Averaging}: \quad \hat{\theta}^{\mathrm{KL}} = \arg\min_{\theta \in \Theta} \sum_k \mathrm{KL}(p(x|\hat{\theta}^k) \,\|\, p(x|\theta)). \tag{1}$$

The estimate $\hat{\theta}^{\mathrm{KL}}$ can also be motivated by a parametric bootstrap procedure that first draws sample $X^{k'}$ from each local model $p(x|\hat{\theta}^k)$, and then estimates a global MLE based on all the combined

bootstrap samples $X' = \{X^{k'}: k \in [d]\}$. We can readily show that this reduces to $\hat{\theta}^{\text{KL}}$ as the size of the bootstrapped samples $X^{k'}$ grows to infinity. Other combination methods based on different distance metrics are also possible, but may not have a similarly natural interpretation.

# 3  Exactness on Full Exponential Families

In this section, we analyze the KL and linear combination methods on full exponential families. We show that the KL combination of the local MLEs exactly equals the global MLE, while the linear average does not in general, but can be made exact by using a special parameterization. This suggests that distributed learning is in some sense "easy" on full exponential families.

**Definition 3.1.** *(1). A family of distributions is said to be a full exponential family if its density can be represented in a canonical form (up to one-to-one transforms of the parameters),*

$$p(x|\theta) = \exp(\theta^T \phi(x) - \log Z(\theta)), \qquad \theta \in \Theta \equiv \{\theta \in \mathbb{R}^m : \int_x \exp(\theta^T \phi(x)) dH(x) < \infty\}.$$

*where $\theta = [\theta_1, \ldots \theta_m]^T$ and $\phi(x) = [\phi_1(x), \ldots \phi_m(x)]^T$ are called the natural parameters and the natural sufficient statistics, respectively. The quantity $Z(\theta)$ is the normalization constant, and $H(x)$ is the reference measure. An exponential family is said to be* minimal *if $[1, \phi_1(x), \ldots \phi_m(x)]^T$ is linearly independent, that is, there is no non-zero constant vector $\alpha$, such that $\alpha^T \phi(x) = 0$ for all $x$.*

**Theorem 3.2.** *If $\mathcal{P} = \{p(x|\theta): \theta \in \Theta\}$ is a full exponential family, then the KL-average $\hat{\theta}^{\text{KL}}$ always exactly recovers the global MLE, that is, $\hat{\theta}^{\text{KL}} = \hat{\theta}^{\text{mle}}$. Further, if $\mathcal{P}$ is minimal, we have*

$$\hat{\theta}^{\text{KL}} = \mu^{-1}\left(\frac{\mu(\hat{\theta}^1) + \cdots + \mu(\hat{\theta}^d)}{d}\right), \tag{2}$$

*where $\mu : \theta \mapsto \mathbb{E}_\theta[\phi(x)]$ is the one-to-one map from the natural parameters to the moment parameters, and $\mu^{-1}$ is the inverse map of $\mu$. Note that we have $\mu(\theta) = \partial \log Z(\theta)/\partial \theta$.*

*Proof.* Directly verify that the KL objective in (1) equals the global negative log-likelihood. $\square$

The nonlinear average in (2) gives an intuitive interpretation of why $\hat{\theta}^{\text{KL}}$ equals $\hat{\theta}^{\text{mle}}$ on full exponential families: it first calculates the local empirical moment parameters $\mu(\hat{\theta}^k) = d/n \sum_{i \in \alpha^k} \phi(x^k)$; averaging them gives the empirical moment parameter on the whole data $\hat{\mu}_n = 1/n \sum_{i \in [n]} \phi(x^k)$, which then exactly maps to the global MLE.

Eq (2) also suggests that $\hat{\theta}^{\text{linear}}$ would be exact only if $\mu(\cdot)$ is an identity map. Therefore, one may make $\hat{\theta}^{\text{linear}}$ exact by using the special parameterization $\vartheta = \mu(\theta)$. In contrast, KL-averaging will make this reparameterization automatically ($\mu$ is different on different exponential families). Note that both KL-averaging and global MLE are invariant w.r.t. one-to-one transforms of the parameter $\theta$, but linear averaging is not.

**Example 3.3** (Variance Estimation). *Consider estimating the variance $\sigma^2$ of a zero-mean Gaussian distribution. Let $\hat{s}_k = (d/n)\sum_{i \in \alpha^k}(x^i)^2$ be the empirical variance on the $k$-th sub-sample and $\hat{s} = \sum_k \hat{s}_k/d$ the overall empirical variance. Then, $\hat{\theta}^{\text{linear}}$ would correspond to different power means on $\hat{s}_k$, depending on the choice of parameterization, e.g.,*

|  | $\theta = \sigma^2$ (variance) | $\theta = \sigma$ (standard deviation) | $\theta = \sigma^{-2}$ (precision) |
|---|---|---|---|
| $\hat{\theta}^{\text{linear}}$ | $\frac{1}{d}\sum_k \hat{s}_k$ | $\frac{1}{d}\sum_k (\hat{s}_k)^{1/2}$ | $\frac{1}{d}\sum_k (\hat{s}_k)^{-1}$ |

*where only the linear average of $\hat{s}_k$ (when $\theta = \sigma^2$) matches the overall empirical variance $\hat{s}$ and equals the global MLE. In contrast, $\hat{\theta}^{\text{KL}}$ always corresponds to a linear average of $\hat{s}_k$, equaling the global MLE, regardless of the parameterization.*

# 4 Information Loss in Distributed Learning

The exactness of $\hat{\theta}^{\mathrm{KL}}$ in Theorem 3.2 is due to the beauty (or simplicity) of exponential families. Following Efron's intuition, full exponential families can be viewed as "straight lines" or "linear subspaces" in the space of distributions, while other distribution families correspond to "curved" sets of distributions, whose deviation from full exponential families can be measured by their *statistical curvatures* as defined by Efron (1975). That work shows that statistical curvature is closely related to Fisher and Rao's theory of second order efficiency (Fisher, 1925, Rao, 1963), and represents the minimum information loss when summarizing the data using first order efficient estimators. In this section, we connect this classical theory with the local learning framework, and show that the statistical curvature also represents the minimum asymptotic deviation of arbitrary combinations of the local MLEs to the global MLE, and that this is achieved by the KL combination method, but not in general by the simpler linear combination method.

## 4.1 Curved Exponential Families and Statistical Curvature

We follow the convention in Efron (1975), and illustrate the idea of statistical curvature using *curved* exponential families, which are smooth sub-families of full exponential families. The theory can be naturally extended to more general families (see e.g., Efron, 1975, Kass and Vos, 2011).

**Definition 4.1.** *A family of distributions $\{p(x|\theta): \theta \in \Theta\}$ is said to be a curved exponential family if its density can be represented as*

$$p(x|\theta) = \exp(\eta(\theta)^T \phi(x) - \log Z(\eta(\theta))), \tag{3}$$

*where the dimension of $\theta = [\theta_1, \ldots, \theta_q]$ is assumed to be smaller than that of $\eta = [\eta_1, \ldots, \eta_m]$ and $\phi = [\phi_1, \ldots, \phi_m]$, that is $q < m$.*

*Following Kass and Vos (2011), we assume some regularity conditions for our asymptotic analysis. Assume $\Theta$ is an open set in $\mathbb{R}^q$, and the mapping $\eta: \Theta \to \eta(\Theta)$ is one-to-one and infinitely differentiable, and of rank $q$, meaning that the $q \times m$ matrix $\dot{\eta}(\theta)$ has rank $q$ everywhere. In addition, if a sequence $\{\eta(\theta_i) \in N_0\}$ converges to a point $\eta(\theta_0)$, then $\{\eta_i \in \Theta\}$ must converge to $\phi(\eta_0)$. In geometric terminology, such a map $\eta: \Theta \to \eta(\Theta)$ is called a q-dimensional* embedding *in $\mathbb{R}^m$.*

Obviously, a curved exponential family can be treated as a smooth subset of a full exponential family $p(x|\eta) = \exp(\eta^T \phi(x) - \log Z(\eta))$, with $\eta$ constrained in $\eta(\Theta)$. If $\eta(\theta)$ is a linear function, then the curved exponential family can be rewritten into a full exponential family in lower dimensions; otherwise, $\eta(\theta)$ is a curved subset in the $\eta$-space, whose curvature – its deviation from planes or straight lines – represents its deviation from full exponential families.

Consider the case when $\theta$ is a scalar, and hence $\eta(\theta)$ is a curve; the geometric curvature $\gamma_\theta$ of $\eta(\theta)$ at point $\theta$ is defined to be the reciprocal of the radius of the circle that fits best to $\eta(\theta)$ locally at $\theta$. Therefore, the curvature of a circle of radius $r$ is a constant $1/r$. In general, elementary calculus shows that $\gamma_\theta^2 = (\dot{\eta}_\theta^T \dot{\eta}_\theta)^{-3} (\ddot{\eta}_\theta^T \ddot{\eta}_\theta \cdot \dot{\eta}_\theta^T \dot{\eta}_\theta - (\ddot{\eta}_\theta^T \dot{\eta}_\theta)^2)$. The *statistical curvature* of a curved exponential family is defined similarly, except equipped with an inner product defined via its Fisher information metric.

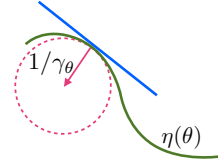

**Definition 4.2** (Statistical Curvature)**.** *Consider a curved exponential family $\mathcal{P} = \{p(x|\theta): \theta \in \Theta\}$, whose parameter $\theta$ is a scalar ($q = 1$). Let $\Sigma_\theta = \mathrm{cov}_\theta[\phi(x)]$ be the $m \times m$ Fisher information on the corresponding full exponential family $p(x|\eta)$. The statistical curvature of $\mathcal{P}$ at $\theta$ is defined as*

$$\gamma_\theta^2 = (\dot{\eta}_\theta^T \Sigma_\theta \dot{\eta}_\theta)^{-3} \big[ (\ddot{\eta}_\theta^T \Sigma_\theta \ddot{\eta}_\theta) \cdot (\dot{\eta}_\theta^T \Sigma_\theta \dot{\eta}_\theta) - (\ddot{\eta}_\theta^T \Sigma_\theta \dot{\eta}_\theta)^2 \big].$$

The definition can be extended to general multi-dimensional parameters, but requires involved notation. We give the full definition and our general results in the appendix.

**Example 4.3** (Bivariate Normal on Ellipse)**.** *Consider a bivariate normal distribution with diagonal covariance matrix and mean vector restricted on an ellipse $\eta(\theta) = [a\cos(\theta), b\sin(\theta)]$, that is,*

$$p(x|\theta) \propto \exp\Big[ -\frac{1}{2}(x_1^2 + x_2^2) + a\cos\theta\, x_1 + b\sin\theta\, x_2 \Big], \quad \theta \in (-\pi, \pi), \ x \in \mathbb{R}^2.$$

*We have that $\Sigma_\theta$ equals the identity matrix in this case, and the statistical curvature equals the geometric curvature of the ellipse in the Euclidian space, $\gamma_\theta = ab(a^2 \sin^2(\theta) + b^2 \cos^2(\theta))^{-3/2}$.*

The statistical curvature was originally defined by Efron (1975) as the minimum amount of information loss when summarizing the sample using first order efficient estimators. Efron (1975) showed that, extending the result of Fisher (1925) and Rao (1963),

$$\lim_{n\to\infty}\left[\mathcal{I}_{\theta^*}^X - \mathcal{I}_{\theta^*}^{\hat{\theta}^{\mathrm{mle}}}\right] = \gamma_{\theta^*}^2 I_{\theta^*}, \tag{4}$$

where $I_{\theta^*}$ is the Fisher information (per data instance) of the distribution $p(x|\theta)$ at the true parameter $\theta^*$, and $\mathcal{I}_{\theta^*}^X = n I_{\theta^*}$ is the total information included in a sample $X$ of size $n$, and $\mathcal{I}_{\theta^*}^{\hat{\theta}^{\mathrm{mle}}}$ is the Fisher information included in $\hat{\theta}^{\mathrm{mle}}$ based on $X$. Intuitively speaking, we lose about $\gamma_{\theta^*}^2$ units of Fisher information when summarizing the data using the ML estimator. Fisher (1925) also interpreted $\gamma_{\theta^*}^2$ as the effective number of data instances lost in MLE, easily seen from rewriting $\mathcal{I}_{\theta^*}^{\hat{\theta}^{\mathrm{mle}}} \approx (n - \gamma_{\theta^*}^2) I_{\theta^*}$, as compared to $\mathcal{I}_{\theta^*}^X = n I_{\theta^*}$. Moreover, this is the minimum possible information loss in the class of "first order efficient" estimators $T(X)$, those which satisfy the weaker condition $\lim_{n\to\infty} \mathcal{I}_{\theta^*}/\mathcal{I}_{\theta^*}^T = 1$. Rao coined the term "second order efficiency" for this property of the MLE.

The intuition here has direct implications for our distributed setting, since $\hat{\theta}^f$ depends on the data only through $\{\hat{\theta}^k\}$, each of which summarizes the data with a loss of $\gamma_{\theta^*}^2$ units of information. The total information loss is $d \cdot \gamma_{\theta^*}^2$, in contrast with the global MLE, which only loses $\gamma_{\theta^*}^2$ overall. Therefore, the additional loss due to the distributed setting is $(d-1) \cdot \gamma_{\theta^*}^2$. We will see that our results in the sequel closely match this intuition.

## 4.2 Lower Bound

The extra information loss $(d-1)\gamma_{\theta^*}^2$ turns out to be the asymptotic lower bound of the mean square error rate $n^2 \mathbb{E}_{\theta^*}[I_{\theta^*}(\hat{\theta}^f - \hat{\theta}^{\mathrm{mle}})^2]$ for any arbitrary combination function $f(\hat{\theta}^1, \dots, \hat{\theta}^d)$.

**Theorem 4.4** (Lower Bound). *For an arbitrary measurable function $\hat{\theta}^f = f(\hat{\theta}^1, \dots, \hat{\theta}^d)$, we have*

$$\liminf_{n\to+\infty} n^2\, \mathbb{E}_{\theta^*}\big[\|f(\hat{\theta}^1, \dots, \hat{\theta}^d) - \hat{\theta}^{\mathrm{mle}}\|^2\big] \ge (d-1)\gamma_{\theta^*}^2 I_{\theta^*}^{-1}.$$

*Sketch of Proof .* Note that

$$
\begin{aligned}
\mathbb{E}_{\theta^*}\big[\|\hat{\theta}^f - \hat{\theta}^{\mathrm{mle}}\|^2\big] &= \mathbb{E}_{\theta^*}\big[\|\hat{\theta}^f - \mathbb{E}_{\theta^*}(\hat{\theta}^{\mathrm{mle}}|\hat{\theta}^1, \dots, \hat{\theta}^d)\|^2\big] + \mathbb{E}_{\theta^*}\big[\|\hat{\theta}^{\mathrm{mle}} - \mathbb{E}_{\theta^*}(\hat{\theta}^{\mathrm{mle}}|\hat{\theta}^1, \dots, \hat{\theta}^d)\|^2\big] \\
&\ge \mathbb{E}_{\theta^*}\big[\|\hat{\theta}^{\mathrm{mle}} - \mathbb{E}_{\theta^*}(\hat{\theta}^{\mathrm{mle}}|\hat{\theta}^1, \dots, \hat{\theta}^d)\|^2\big] \\
&= \mathbb{E}_{\theta^*}\big[\mathrm{var}_{\theta^*}(\hat{\theta}^{\mathrm{mle}}|\hat{\theta}^1, \dots, \hat{\theta}^d)\big],
\end{aligned}
$$

where the lower bound is achieved when $\hat{\theta}^f = \mathbb{E}_{\theta^*}(\hat{\theta}^{\mathrm{mle}}|\hat{\theta}^1, \dots, \hat{\theta}^d)$. The conclusion follows by showing that $\lim_{n\to+\infty} \mathbb{E}_{\theta^*}[\mathrm{var}_{\theta^*}(\hat{\theta}^{\mathrm{mle}}|\hat{\theta}^1, \dots, \hat{\theta}^d)] = (d-1)\gamma_{\theta^*}^2 I_{\theta^*}^{-1}$; this requires involved asymptotic analysis, and is presented in the Appendix. ∎

The proof above highlights a geometric interpretation via the projection of random variables (e.g., Van der Vaart, 2000). Let $\mathcal{F}$ be the set of all random variables in the form of $f(\hat{\theta}^1, \dots, \hat{\theta}^d)$. The optimal consensus function should be the projection of $\hat{\theta}^{\mathrm{mle}}$ onto $\mathcal{F}$, and the minimum mean square error is the distance between $\hat{\theta}^{\mathrm{mle}}$

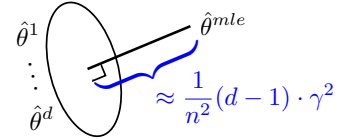

and $\mathcal{F}$. The conditional expectation $\hat{\theta}^f = \mathbb{E}_{\theta^*}(\hat{\theta}^{\mathrm{mle}}|\hat{\theta}^1, \dots, \hat{\theta}^d)$ is the exact projection and ideally the best combination function; however, this is intractable to calculate due to the dependence on the unknown true parameter $\theta^*$. We show in the sequel that $\hat{\theta}^{\mathrm{KL}}$ gives an efficient approximation and achieves the same asymptotic lower bound.

## 4.3 General Consistent Combination

We now analyze the performance of a general class of $\hat{\theta}^f$, which includes both the KL average $\hat{\theta}^{\mathrm{KL}}$ and the linear average $\hat{\theta}^{\mathrm{linear}}$; we show that $\hat{\theta}^{\mathrm{KL}}$ matches the lower bound in Theorem 4.4, while $\hat{\theta}^{\mathrm{linear}}$ is not optimal even on full exponential families. We start by defining conditions which any "reasonable" $f(\hat{\theta}^1, \dots, \hat{\theta}^d)$ should satisfy.

**Definition 4.5.** *(1).  We say $f(\cdot)$ is* consistent, *if for $\forall \theta \in \Theta$, $\theta^k \to \theta$, $\forall k \in [d]$ implies $f(\theta^1, \ldots, \theta^d) \to \theta$.*

*(2). $f(\cdot)$ is* symmetric *if $f(\hat{\theta}^1, \ldots, \hat{\theta}^d) = f(\hat{\theta}^{\sigma(1)}, \ldots, \hat{\theta}^{\sigma(d)})$, for any permutation $\sigma$ on $[d]$.*

The consistency condition guarantees that if all the $\hat{\theta}^k$ are consistent estimators, then $\hat{\theta}^f$ should also be consistent. The symmetry is also straightforward due to the symmetry of the data partition $\{X^k\}$. In fact, if $f(\cdot)$ is not symmetric, one can always construct a symmetric version that performs better or at least the same (see Appendix for details). We are now ready to present the main result.

**Theorem 4.6.** *(1).  Consider a consistent and symmetric $\hat{\theta}^f = f(\hat{\theta}^1, \ldots, \hat{\theta}^d)$ as in Definition 4.5, whose first three orders of derivatives exist. Then, for curved exponential families in Definition 4.1,*

$$\mathbb{E}_{\theta^*}[\hat{\theta}^f - \hat{\theta}^{\mathrm{mle}}] = \frac{d-1}{n}\beta_{\theta^*}^f + o(n^{-1}),$$

$$\mathbb{E}_{\theta^*}[\|\hat{\theta}^f - \hat{\theta}^{\mathrm{mle}}\|^2] = \frac{d-1}{n^2} \cdot [\gamma_{\theta^*}^2 I_{\theta^*}^{-1} + (d+1)(\beta_{\theta^*}^f)^2] + o(n^{-2}),$$

*where $\beta_{\theta^*}^f$ is a term that depends on the choice of the combination function $f(\cdot)$. Note that the mean square error is consistent with the lower bound in Theorem 4.4, and is tight if $\beta_{\theta^*}^f = 0$.*

*(2). The KL average $\hat{\theta}^{\mathrm{KL}}$ has $\beta_{\theta^*}^f = 0$, and hence achieves the minimum bias and mean square error,*

$$\mathbb{E}_{\theta^*}[\hat{\theta}^{\mathrm{KL}} - \hat{\theta}^{\mathrm{mle}}] = o(n^{-1}), \qquad \mathbb{E}_{\theta^*}[\|\hat{\theta}^{\mathrm{KL}} - \hat{\theta}^{\mathrm{mle}}\|^2] = \frac{d-1}{n^2} \cdot \gamma_{\theta^*}^2 I_{\theta^*}^{-1} + o(n^{-2}).$$

*In particular, note that the bias of $\hat{\theta}^{\mathrm{KL}}$ is smaller in magnitude than that of general $\hat{\theta}^f$ with $\beta_{\theta^*}^f \neq 0$.*

*(4). The linear averaging $\hat{\theta}^{\mathrm{linear}}$, however, does not achieve the lower bound in general. We have*

$$\beta_{\theta^*}^{\mathrm{linear}} = I_*^{-2}(\ddot{\eta}_{\theta^*}^T \Sigma_{\theta^*} \dot{\eta}_{\theta^*} + \frac{1}{2}\mathbb{E}_{\theta^*}\Big[\frac{\partial^3 \log p(x|\theta^*)}{\partial \theta^3}\Big]),$$

*which is in general non-zero even for full exponential families.*

*(5). The MSE w.r.t. the global MLE $\hat{\theta}^{\mathrm{mle}}$ can be related to the MSE w.r.t. the true parameter $\theta^*$, by*

$$\mathbb{E}_{\theta^*}[\|\hat{\theta}^{\mathrm{KL}} - \theta^*\|^2] = \mathbb{E}_{\theta^*}[\|\hat{\theta}^{\mathrm{mle}} - \theta^*\|^2] + \frac{d-1}{n^2} \cdot \gamma_{\theta^*}^2 I_{\theta^*}^{-1} + o(n^{-2}).$$

$$\mathbb{E}_{\theta^*}[\|\hat{\theta}^{\mathrm{linear}} - \theta^*\|^2] = \mathbb{E}_{\theta^*}[\|\hat{\theta}^{\mathrm{mle}} - \theta^*\|^2] + \frac{d-1}{n^2} \cdot [\gamma_{\theta^*}^2 I_{\theta^*}^{-1} + 2(\beta_{\theta^*}^{\mathrm{linear}})^2] + o(n^{-2}).$$

*Proof.* See Appendix for the proof and the general results for multi-dimensional parameters. $\square$

Theorem 4.6 suggests that $\hat{\theta}^f - \hat{\theta}^{\mathrm{mle}} = O_p(1/n)$ for any consistent $f(\cdot)$, which is smaller in magnitude than $\hat{\theta}^{\mathrm{mle}} - \theta^* = O_p(1/\sqrt{n})$. Therefore, any consistent $\hat{\theta}^f$ is first order efficient, in that its difference from the global MLE $\hat{\theta}^{\mathrm{mle}}$ is negligible compared to $\hat{\theta}^{\mathrm{mle}} - \theta^*$ asymptotically. This also suggests that KL and the linear methods perform roughly the same asymptotically in terms of recovering the true parameter $\theta^*$. However, we need to treat this claim with caution, because, as we demonstrate empirically, the linear method may significantly degenerate in the non-asymptotic region or when the conditions in Theorem 4.6 do not hold.

## 5  Experiments and Practical Issues

We present numerical experiments to demonstrate the correctness of our theoretical analysis. More importantly, we also study empirical properties of the linear and KL combination methods that are not enlightened by the asymptotic analysis. We find that the linear average tends to degrade significantly when its local models ($\hat{\theta}^k$) are not already close, for example due to small sample sizes, heterogenous data partitions, or non-convex likelihoods (so that different local models find different local optima). In contrast, the KL combination is much more robust in practice.

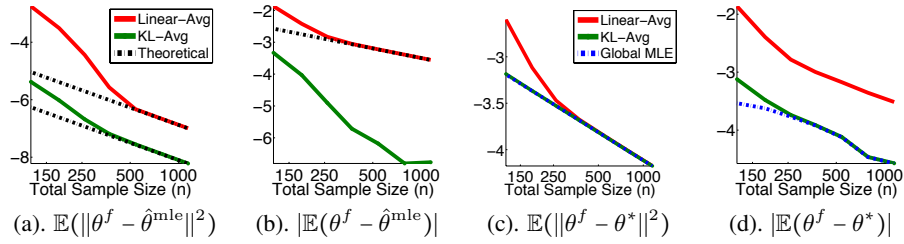

(a). $\mathbb{E}(\|\theta^f - \hat{\theta}^{\mathrm{mle}}\|^2)$   (b). $|\mathbb{E}(\theta^f - \hat{\theta}^{\mathrm{mle}})|$   (c). $\mathbb{E}(\|\theta^f - \theta^*\|^2)$   (d). $|\mathbb{E}(\theta^f - \theta^*)|$

Figure 1: Result on the toy model in Example 4.3. (a)-(d) The mean square errors and biases of the linear average $\hat{\theta}^{\mathrm{linear}}$ and the KL average $\hat{\theta}^{\mathrm{KL}}$ w.r.t. to the global MLE $\hat{\theta}^{\mathrm{mle}}$ and the true parameter $\theta^*$, respectively. The y-axes are shown on logarithmic (base 10) scales.

## 5.1   Bivariate Normal on Ellipse

We start with the toy model in Example 4.3 to verify our theoretical results. We draw samples from the true model (assuming $\theta^* = \pi/4$, $a = 1$, $b = 5$), and partition the samples randomly into 10 subgroups ($d = 10$). Fig. 1 shows that the empirical biases and MSEs match closely with the theoretical predictions when the sample size is large (e.g., $n \geq 250$), and $\hat{\theta}^{\mathrm{KL}}$ is consistently better than $\hat{\theta}^{\mathrm{linear}}$ in terms of recovering both the global MLE and the true parameters. Fig. 1(b) shows that the bias of $\hat{\theta}^{\mathrm{KL}}$ decreases faster than that of $\hat{\theta}^{\mathrm{linear}}$, as predicted in Theorem 4.6 (2). Fig. 1(c) shows that all algorithms perform similarly in terms of the asymptotic MSE w.r.t. the true parameters $\theta^*$, but linear average degrades significantly in the non-asymptotic region (e.g., $n < 250$).

**Model Misspecification.** Model misspecification is unavoidable in practice, and may create multiple local modes in the likelihood objective, leading to poor behavior from the linear average. We illustrate this phenomenon using the toy model in Example 4.3, assuming the true model is $\mathcal{N}([0, 1/2], \mathbf{1}_{2\times2})$, outside of the assumed parametric family. This is illustrated in the figure at right, where the ellipse represents the parametric family, and the black square denotes the true model. The MLE will concentrate on the projection of the true model to the ellipse, in one of two locations ($\theta = \pm\pi/2$) indicated by the two red circles. Depending on the random data sample, the global MLE will concentrate on one or the other of these two values; see Fig. 2(a). Given a sufficient number of samples ($n > 250$), the probability that the MLE is at $\theta \approx -\pi/2$ (the less favorable mode) goes to zero. Fig. 2(b) shows KL averaging mimics the bi-modal distribution of the global MLE across data samples; the less likely mode vanishes slightly slower. In contrast, the linear average takes the arithmetic average of local models from both of these two local modes, giving unreasonable parameter estimates that are close to neither (Fig. 2(c)).

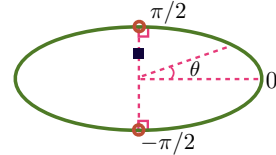

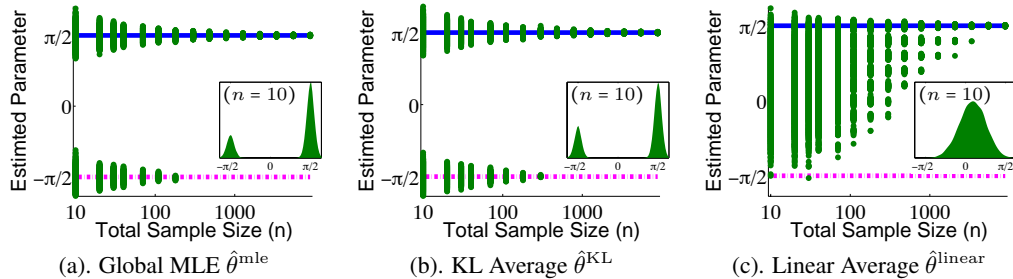

(a). Global MLE $\hat{\theta}^{\mathrm{mle}}$   (b). KL Average $\hat{\theta}^{\mathrm{KL}}$   (c). Linear Average $\hat{\theta}^{\mathrm{linear}}$

Figure 2: Result on the toy model in Example 4.3 with model misspecification: scatter plots of the estimated parameters vs. the total sample size $n$ (with 10,000 random trials for each fixed $n$). The inside figures are the densities of the estimated parameters with fixed $n = 10$. Both global MLE and KL-average concentrate on two locations ($\pm\pi/2$), and the less favorable ($-\pi/2$) vanishes when the sample sizes are large (e.g., $n > 250$). In contrast, the linear approach averages local MLEs from the two modes, giving unreasonable estimates spread across the full interval.

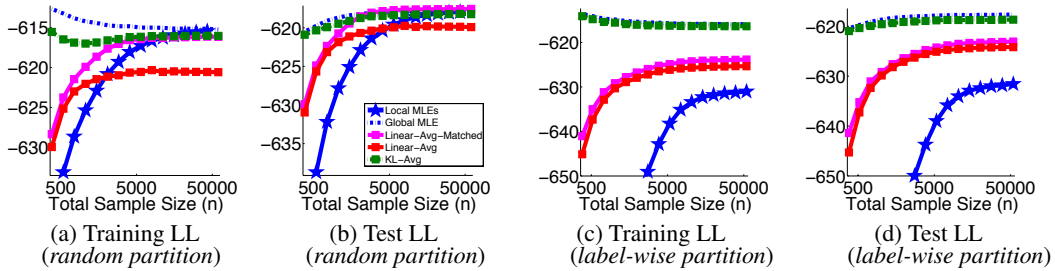

| (a) Training LL | (b) Test LL | (c) Training LL | (d) Test LL |
| *(random partition)* | *(random partition)* | *(label-wise partition)* | *(label-wise partition)* |

Figure 3: Learning Gaussian mixture models on MNIST: training and test log-likelihoods of different methods with varying training size $n$. In (a)-(b), the data are partitioned into 10 sub-groups uniformly at random (ensuring sub-samples are i.i.d.); in (c)-(d) the data are partitioned according to their digit labels. The number of mixture components is fixed to be 10.

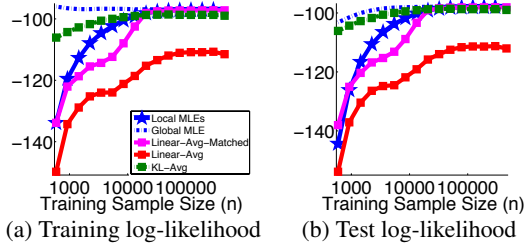

(a) Training log-likelihood   (b) Test log-likelihood

Figure 4: Learning Gaussian mixture models on the YearPredictionMSD data set. The data are randomly partitioned into 10 sub-groups, and we use 10 mixture components.

## 5.2 Gaussian Mixture Models on Real Datasets

We next consider learning Gaussian mixture models. Because component indexes may be arbitrarily switched, naïve linear averaging is problematic; we consider a *matched linear average* that first matches indices by minimizing the sum of the symmetric KL divergences of the different mixture components. The KL average is also difficult to calculate exactly, since the KL divergence between Gaussian mixtures is intractable. We approximate the KL average using Monte Carlo sampling (with 500 samples per local model), corresponding to the parametric bootstrap discussed in Section 2.

We experiment on the MNIST dataset and the YearPredictionMSD dataset in the UCI repository, where the training data is partitioned into 10 sub-groups randomly and evenly. In both cases, we use the original training/test split; we use the full testing set, and vary the number of training examples $n$ by randomly sub-sampling from the full training set (averaging over 100 trials). We take the first 100 principal components when using MNIST. Fig. 3(a)-(b) and 4(a)-(b) show the training and test likelihoods. As a baseline, we also show the average of the log-likelihoods of the local models (marked as `local MLEs` in the figures); this corresponds to randomly selecting a local model as the combined model. We see that the KL average tends to perform as well as the global MLE, and remains stable even with small sample sizes. The naïve linear average performs badly even with large sample sizes. The matched linear average performs as badly as the naïve linear average when the sample size is small, but improves towards to the global MLE as sample size increases.

For MNIST, we also consider a severely heterogenous data partition by splitting the images into 10 groups according to their digit labels. In this setup, each partition learns a local model only over its own digit, with no information about the other digits. Fig. 3(c)-(d) shows the KL average still performs as well as the global MLE, but both the naïve and matched linear average are much worse even with large sample sizes, due to the dissimilarity in the local models.

## 6 Conclusion and Future Directions

We study communication-efficient algorithms for learning generative models with distributed data. Analyzing both a common linear averaging technique and a less common KL-averaging technique provides both theoretical and empirical insights. Our analysis opens many important future directions, including extensions to high dimensional inference and efficient approximations for complex machine learning models, such as LDA and neural networks.

**Acknowledgements.**   This work sponsored in part by NSF grants IIS-1065618 and IIS-1254071, and the US Air Force under Contract No. FA8750-14-C-0011 under DARPA's PPAML program.

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
