[Reviews · NeurIPS 2014]

Submitted by Assigned_Reviewer_16

This paper studies a two-step procedure for distributed learning that follows a “divide and conquer” strategy. The author analyze the asymptotic property and information loss bound of the linear averaging and KL-Averaging method. They point out that KL-Averaging method outperforms the Linear-Averaging in fully and curved exponential distributions which agrees with the intuition. The notations of this paper are clear and consistent. However, this paper does not consider the computational cost, which could be expensive by using such a KL-Averaging method, which is critical by using a “divide and conquer” framework.
Summary: This paper shows that the KL-Averaging methods generally outperforms Linear-Averaging and provides theory to support the KL-Averaging methods. However, this paper does not consider the computational cost.

Submitted by Assigned_Reviewer_25

Paper summary:
The paper analyzes methods for combining maximum-likelihood estimates obtained from different partitions of a training data set, such as linear averaging and the KL-divergence based averaging introduced by Merugu and Ghosh (ICDM, 2003). It applies Statistical Curvature theory to show that the latter, among all symmetric and consistent combination methods, achieves the minimum mean square error, while the linear averaging - in general - doesn't. Experiments provide further insights into the robustness of both methods on finite samples and real-world data.

Quality/Clarity:
The paper doesn't follow all the NIPS formatting requirements (missing line numbers; citations within the text should be numbered consecutively in squared brackets). The technical quality appears solid (although I didn't check the proofs in the Supplementary material), and the presentation is clear. A strong point are the examples and small illustrations which help the reader to develop intuition. Only a few minor remarks:
- \alpha_k is sometimes written with sub-, sometimes with super-scripts (when appearing as summation index set).
- In Definition 4.1, I didn't understand the conditions related to the sequence \eta(\theta_i). In particular, what is N_0 here, and what is the relation between \eta(\theta_i) and \eta_i?
- The citation style (page 9) isn't always consistent, e.g., abbrevation of authors' first names

Originality/Significance:
From my impression, the application of Statistical Curvature theory for a rigorous analysis of combination methods is an original and elegant idea. The main result in Theorem 4.6 is significant also for practitioners. I believe this paper would be of interest for a large part of the NIPS community and inspire more work in this direction.
Summary: Summary of review:
+ Relevant problem for the NIPS community, original approach with significant results
+ Technically sound, well-written and -organized
+ Strong theoretical results with implications also for practitioners

Submitted by Assigned_Reviewer_42

The main message of this paper is: IF you want to perform density/parameter estimation for an exponential family by dividing the data then combining the locally optimal estimators, THEN use KL as the combining function BECAUSE 1) if the distribution is a full exponential family then you get the globally optimal solution and 2) if the distribution is a curved exponential family then this the best you can do. The rest of the paper proves these results and showed some empirically evidence on toy datasets.

The main message of this paper is theoretical however it still didn't answer a few questions:

1- Why do I have to follow the divide and combine approach? Granted KL is the best if this is all what I can do, but what if I can use asynchronous techniques (or synchronous algorithms) to perform distributed inference? what are the trade-offs?

2- At least I would have liked to see point 1 addressed experimentally.

3- It would be great to link the results in this paper with the following paper:

Parallelized Stochastic Gradient Descent
http://martin.zinkevich.org/publications/nips2010.pdf

It addresses the same setup from an optimization point of view and showed the optimality of the linear averaging method under certain conditions. Can you relate these two views? Take for instance logistic regression that is also a member of the exponential family, what can you say here?

4 - From a practical point of view, even using the method in the paper cited above, there is still a gap between theory and practice. From a hands-on experience on very large dataset (that is hundreds of millions of examples and tens of millions of features), performing divide and combine (via linear average) in a simple SGD setting for even linear/logistic regression results in a solution that is equivalent to the global solution over 5-15% less data (i.e. there is a loss of information). It is not clear to me if this is because of the linear averaging or something more sophisticated can be said using the KL method instead.

5 - I understand that this is a theory paper but the experimental section could be stronger. What happens when you vary the number of features, number of groups, Number of model parameters (# of mixtures)?

- please mention in the paper under what family (curved or full) is the mixture of Gaussian example?

Summary: Interesting theoretical results but the practical implication is not clear to me at least given the simple models analyzed in the paper. There are interesting future directions though.
Author Feedback
Author rebuttal: We thank both reviewers for their comments, and will fix the format and typo issues in the final version.

R_16: The computational cost/method varies from model to model, and should be established case by case. In the paper, we used the bootstrap procedure discussed in Section 2 (a Monte Carlo approximation) for the case of Gaussian mixture on real datasets, and find it works efficiently.

Note that our framework uses only a one-shot exchange of information, compared to many divide-and-conquer methods that use iterative updates (e.g., ADMM).